# In vivo study of gene expression with an enhanced dual-color fluorescent transcriptional timer

Li He[1]*, Richard Binari[1,2], Jiuhong Huang[3], Julia Falo-Sanjuan[4], Norbert Perrimon[1,2]*

[1]Department of Genetics, Harvard Medical School, Boston, United States; [2]Howard Hughes Medical Institute, Boston, United States; [3]International Academy of Targeted Therapeutics and Innovation, Chongqing University of Arts and Sciences, Chongqing, China; [4]Tufts University, Medford, United States

**Abstract** Fluorescent transcriptional reporters are widely used as signaling reporters and biomarkers to monitor pathway activities and determine cell type identities. However, a large amount of dynamic information is lost due to the long half-life of the fluorescent proteins. To better detect dynamics, fluorescent transcriptional reporters can be destabilized to shorten their half-lives. However, applications of this approach in vivo are limited due to significant reduction of signal intensities. To overcome this limitation, we enhanced translation of a destabilized fluorescent protein and demonstrate the advantages of this approach by characterizing spatio-temporal changes of transcriptional activities in *Drosophila*. In addition, by combining a fast-folding destabilized fluorescent protein and a slow-folding long-lived fluorescent protein, we generated a dual-color transcriptional timer that provides spatio-temporal information about signaling pathway activities. Finally, we demonstrate the use of this transcriptional timer to identify new genes with dynamic expression patterns.

DOI: https://doi.org/10.7554/eLife.46181.001

*For correspondence:
Li_He@hms.harvard.edu (LH);
perrimon@receptor.med.harvard.edu (NP)

**Competing interests:** The authors declare that no competing interests exist.

## Introduction

Changes in gene expression are one of the key mechanisms that organisms use during both development and homeostasis. Gene expression is a highly dynamic process, which not only bears critical information about regulatory mechanisms but also controls the fate of many biological processes (*Purvis and Lahav, 2013*; *Yosef and Regev, 2011*). For example, oscillatory or constant expression of the Notch effector *Hes1* dictates the choice of neuron stem cells between proliferation and differentiation (*Isomura and Kageyama, 2014*). In addition, defining the exact 'on' and 'off' timing of a relevant signal is vital to control different developmental events (*Doupé and Perrimon, 2014*). For example, during the development of fly compound eyes, simultaneous activation of EGF and Notch signals determines a cone cell fate (*Flores et al., 2000*), while cells that experience sequential expression of EGF and the Notch-ligand Delta differentiate into photoreceptor cells (*Tsuda et al., 2002*).

Documenting precisely the spatio-temporal changes in gene expression that occur in response to intrinsic and extrinsic signals is a challenging problem in cell and developmental biology. Traditionally, transcriptional reporters that drive expression of fluorescent proteins (FPs) under the control of signaling response elements (SREs) have been widely used to visualize the activities of transcriptional events; however, the slow degradation (half-life >20 hr) of FPs makes it hard to achieve the temporal resolution needed to dissect the dynamic nature of gene expression. Recently, this problem has been addressed by the application of a fluorescent timer, a slow maturing fluorescent protein that

**eLife digest** Fruit flies and other animals have complex body plans containing many different types of cells. To make and maintain these body plans, individual genes must be switched on and off at specific times in particular cells to control how the animal grows. Some of these genes may be switched on for long periods of time, while others may be rapidly switched on and off on repeated occasions.

Fluorescent reporter proteins have been extensively used to study gene activity in cells. Typically, this involves linking the gene encoding the fluorescent reporter to a gene of interest, so that when the gene is switched on a fluorescent protein will also be produced. The fluorescent protein emits light of a particular color and measuring this light provides a way to monitor a gene's activity.

Unfortunately, fluorescent proteins tend to break down slowly, and the level of fluorescence emitted cannot fluctuate quickly enough to reflect rapid changes in gene behavior. One way to overcome this limitation is to use destabilized fluorescent proteins that degrade more rapidly inside cells. However, current strategies for creating these proteins cause them to emit less light, making fluorescence more difficult to detect.

To address this issue, He et al. developed a new green destabilized protein, adding elements that increase production of the protein so a greater amount of light can be emitted. The green destabilized protein was then combined with a red fluorescent reporter that degrades more slowly to develop a new tool called TransTimer. When the gene linked to the reporter switches on, the green destabilized protein turns on before the red reporter turns on. But, as the gene switches off, the destabilized protein will degrade until only the red signal remains. This allows the ratio of green to red color emitted from the TransTimer to indicate the timing of a gene's activity.

Using this tool, He et al. uncovered new details about the patterns of activity of two signals, known as Notch and STAT, that were largely missed by studies using traditional fluorescent reporters. Further experiments demonstrated that TransTimer can be used to carry out large-scale screens in living fruit flies, which have not been possible with more time consuming live-cell imaging techniques.

The fluorescent reporter developed by He et al. will be a useful tool to understand when and where genes are switched on during the lives of fruit flies. In the future, TransTimer could be adapted for use in other model animals or plants.

DOI: https://doi.org/10.7554/eLife.46181.002

changes its from blue to red in ~7 hr (*Bending et al., 2018*). Despite the still relatively long conversion time, this fluorescent timer has two additional limitations: the signal is hard to fix for long-term storage, and because it can be photoconverted from blue to red, this timer only allows one single image and prohibits live-imaging application (*Subach et al., 2009*). Another strategy is the development of a destabilized version of GFP with a half-life of ~2 hr, which is achieved by fusing GFP with a PEST peptide signal for protein degradation (*Li et al., 1998*; *Rogers et al., 1986*). However, despite many in vitro successes, this strategy has met a major limitation when applied in vivo due to substantial loss of fluorescent intensity. Therefore, regular stable FPs are still the primary choice for generating transcriptional reporters to study gene expression patterns in vivo.

Here, we address this problem by using translational enhancers to boost production of the destabilized reporters and demonstrate the advantages of using short-lived FPs to study dynamic gene expression in vivo. In addition, we generate a transcriptional timer that can be readily applied to study spatio-temporal activation of signaling pathways. Finally, we document how this transcriptional timer can be used, either using the UAS/Gal4 system or in an enhancer trap screen, to identify genes with dynamic expression.

## Results

### Current limitations of stable transcriptional reporters and challenges of the destabilization strategy

Matching reporter dynamics with the activity of target genes is essential to faithfully recapitulate signaling activities (*Doupé and Perrimon, 2014*). Two primary kinetic properties dictate reporter activities: the 'switch-on' and 'switch-off' speeds. The 'on' kinetic of FPs have been improved by engineering fast folding FPs, which shorten the maturation time of FPs from more than 1 hr to less than 10min (*Pédelacq et al., 2006*) (*Gordon et al., 2007*). The 'off' kinetics of FPs has been improved from more than 20 hr to around 2 hr by fusing FPs with PEST peptides that promote degradation (*Li et al., 1998*) (*Figure 1a*). Although the advantages of using a short-lived reporter have been previously reported (*Li et al., 1998*), a systematic analysis of the differences between long-lived and short-lived reporters is still lacking. Using a protein synthesis and degradation model (*Figure 1—figure supplement 1a–c*), we first simulated the dynamics of the reporters and demonstrated a significant improvement by decreasing the half-life ($Tp_{1/2}$) of the reporters from 20 hr to 2 hr (*Figure 1b–f*). Specifically, we illustrate this problem using simulated responses of FP reporters to four basic types of promoter activities: switch on, switch off, pulse activation, and oscillation. Compared to FPs with a half-life of 20 hr, FPs with a half-life of 2 hr have a 90% shorter response time (time to achieve 50% maximal intensity) during the 'on' or 'off' events, and up to four times larger dynamic range in the case of oscillatory expression (*Figure 1e,f*, *Figure 1—figure supplement 1a–c*).

Although the destabilization strategy successfully leads to an FP with a shorter half-life, it is problematic as it causes significant loss of the signal (*Figure 1g–i*). Thus, at constant expression, the intensity of the maximum signal is linearly proportional to the protein half-life, and decreasing the half-life from 20 hr to 2 hr causes a 90% signal loss (*Figure 1g*, black curve). The reduction of maximum signal intensity is also affected by different types of transcriptional activation. In the case of short pulsatile activation, a more transient activation (a shorter duration Td) is less sensitive to a reduction in the protein half-life (*Figure 1g* colored curves); however, transient activation also triggers weaker reporter activity, which makes it more vulnerable to intensity reduction.

We further analyzed the effects of reducing mRNA half-life ($Tm_{1/2}$) compared to protein destabilization (*Figure 1h,i*). Measurements of $Tm_{1/2}$ of FPs in the literature are highly variable, ranging from several minutes to hours (*Baker and Parker, 2006*; *Sacchetti et al., 2001*; *Houser et al., 2012*), which is probably due to differences in the 3' UTR used or different mRNA expression levels relative to the mRNA degradation machinery. According to our measurements (*Figure 2—figure supplement 1e*), the $Tm_{1/2}$ of FP reporters is about 0.5 hr. Therefore, we used 0.5 hr to 3 hr in our modeling. According to the model, the mRNA half-life significantly influences reporter intensity (*Figure 1h*), which is approximately proportional to the $Tp_{1/2}* Tm_{1/2}$. In contrast, reducing mRNA half-life has much less effect on reporter response time, which is mainly controlled by $Tp_{1/2} + Tm_{1/2}$ (*Figure 1i*). Because in our system, $Tp_{1/2}$ is much longer than $Tm_{1/2}$, shortening the mRNA lifetime will significantly reduce signal intensity without endowing the reporter with more range in dynamic detection. Therefore, we decided to primarily use destabilized FPs in our study. For a system with a large $Tm_{1/2}$ relative to $Tp_{1/2}$, strategies to shorten mRNA lifetime by addition of RNA destabilizing sequence can be used (*Voon et al., 2005*).

### Application of translational enhancers to rescue the signal loss caused by destabilization

The significant loss of signal limits implementation of the destabilization strategy, especially in systems like *Drosophila* where a transgene is usually present in 1 or 2 copies per genome. To overcome this obstacle, we searched for ways to increase the signal of destabilized FPs. One possible solution is to use FPs with high intrinsic brightness. To test this, we used a fly codon-optimized sfGFP for its fast folding and bright fluorescence (*Pédelacq et al., 2006*; *Venken et al., 2011*). The effectiveness of destabilization was first tested in cultured fly S2 cells. Adding the PEST sequence from mouse ornithine decarboxylase (MODC) effectively reduced the half-life of sfGFP to ~3 hr (*Figure 2—figure supplement 1*). Next, to test the approach in vivo, we generated transgenic flies with destabilized sfGFP (dGFP) for two widely used signaling reporters: STAT (containing the STAT response element

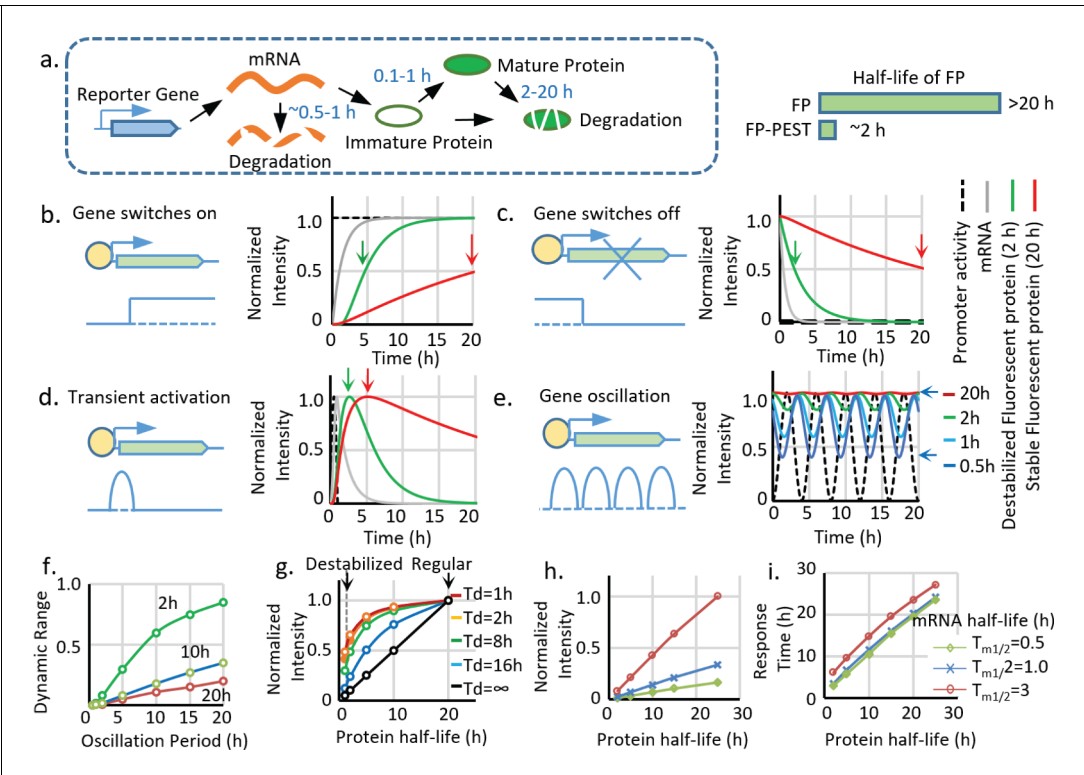

**Figure 1.** Advantages and limitations of destabilized fluorescent transcriptional reporters. (a) Illustration of the biological processes that affect the final concentration of mature fluorescent protein, including transcription, translation, protein maturation, mRNA degradation, and protein degradation with the general half-life of mRNA, FP maturation time, and half-life of protein labeled. Destabiliation of the FP, achieved by fusion of FP with PEST domain, shortens the half-life of regular FP from over ~20 hr to ~2 hr. (b-d). Comparison between simulated signals from a destabilized fluorescent reporter (green, $T_{p1/2}$ = 2 hr) and a regular fluorescent reporter (blue, $T_{p1/2}$ = 20 hr) following switch-on, switch-off, 1 hr pulse, and oscillation with a period of 2 hr. The half-life of mRNA ($T_{m1/2}$) is set as 0.5 hr, and the protein maturation time ($\tau_{m1/2}$) is set as 0.1 hr. The time points when the fluorescent signals reach 50% of the maximal intensity (for switch-on and switch-off) and the maximum response (for transient pulse activation) are indicated by black arrows. (e). Simulated signal intensities of fluorescent proteins with different protein half-lives to a sinusoid transcriptional oscillation with a period of 4 hr. The rest of the parameters are the same with above. The dynamic portion (the difference between the peak and valley) of the reporter (a half-life of 0.5 hr) is indicated by black arrows. (f). The dynamic range (the difference between the peak and valley of the signal compared to its average intensity as indicated in (e) of reporters with indicated half-lives generated by sinusoid transcriptional activity with different length of the period. The dynamic range of the total signal positively correlates with the oscillation period and negatively correlates with the protein half-life. (g) The intensity of the reporter signal for a constitutively active promoter (with no temporal variation) linearly depends on its protein half-life (black line). However, the maximal intensity of the reporter is less reduced by the shortened half-life for a shorter transient activation: a short-lived reporter (half-life of 2 hr) shows a 90% reduction of maximal signal compared with a stable reporter (half-life of 20 hr) for a constitutively active promoter. Nevertheless, the intensity is only reduced by 50% if the promoter is transiently activated for 1 hr. Td: the duration of a pulse transient promoter activation. (h,i) Simulated changes in maximum fluorescent intensity and response time (time to reach 50% of the maximum intensity) of the reporters with different half-lives of protein and mRNA for a promoter switch-off event.

DOI: https://doi.org/10.7554/eLife.46181.003

The following figure supplement is available for figure 1:

**Figure supplement 1.** Model of the transcriptional reporter synthesis, maturation, and degradation.

DOI: https://doi.org/10.7554/eLife.46181.004

from *Socs36E*; *Bach et al., 2007*) and Notch (containing the Notch response element Su(H)Gbe; *Furriols and Bray, 2001*). GFP signals were examined in tissues previously reported to show high STAT and Notch activities (embryo for STAT and wing imaginal disc for Notch). Destabilization reduced the signal intensity of these reporters to near background (*Figure 2a–c*). As further increasing the intrinsic brightness of FPs is challenging—even with the brightest FPs currently available the increase in signal intensity is still limited (less than two fold; *Cranfill et al., 2016*)—we decided to test other strategies to increase the FP signal. One potential solution is to increase expression of the FP by expressing multiple tandem FPs (*Shearin et al., 2014*; *Genové et al., 2005*). However, this

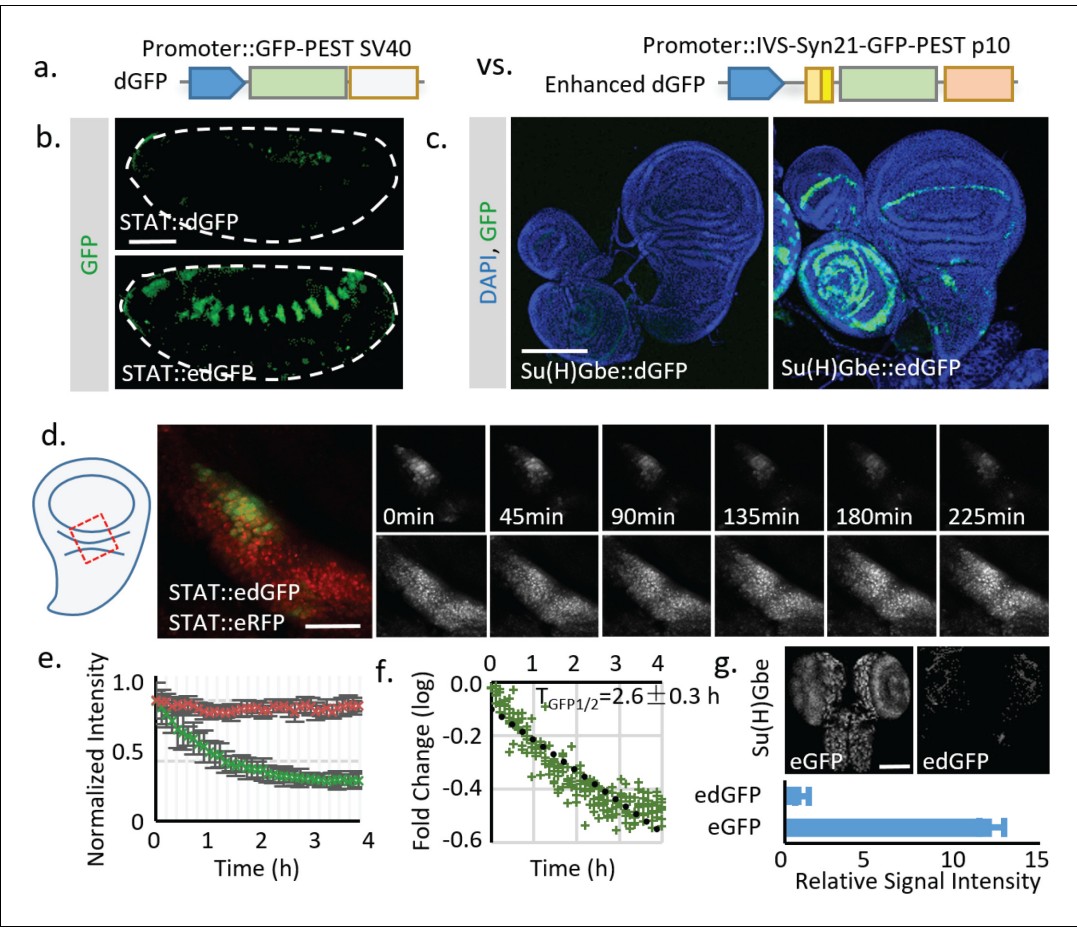

**Figure 2.** Using translational enhancers to increase the signal from destabilized fluorescent reporters for in vivo study. (a) Illustration of the regular and destabilized GFP reporters. All the GFPs used in this study are the fast folding superfolder GFP (sfGFP). Destabilized GFP is labeled as dGFP, and dGFP reporter with translational enhancing elements is labeled as edGFP. All FPs used in this study contain the SV40 nuclear localization signal (NLS) at the N-terminus to facilitate signal segregation unless specified otherwise. (b, c) Comparison of dGFP and edGFP controlled by 6XSTAT response element in fly embryos and Su(H)Gbe Notch responding element in third instar wing imaginal discs. Images were taken with identical exposure. The contour of the embryo is outlined. (d) Dissected fly wing disc, expressing the *STAT::edGFP* and *STAT:: eRFP*, cultured ex vivo. Tissue was treated with 10 µM Actinomycin D to block transcription. STAT at the hinge region of the wing disc was imaged every 5 min for 4.5 hr. (e) The intensities of both dGFP and RFP were measured over time. Data from three independent replicates were collected and plotted. (f) The in vivo reporter half-life ($T_{GFP1/2}$, representing effects of both $T_{m1/2}$ and $T_{p1/2}$) was estimated by linear regression of fluorescent intensity (in logarithmic scale). 95% confidence interval was calculated from linear regression. (g) Regular GFP and dGFP are expressed under Su(H)Gbe together with translational enhancers. Images were taken under identical parameters. The total fluorescent intensity from both reporters was plotted below with the intensity normalized to the dGFP signal. Data were collected from 10 different brains for each genotype. Scale bar: (b) 50 µm; (c, g) 100 µm; (d) 25 µm. Error bar: s.e.m.

DOI: https://doi.org/10.7554/eLife.46181.005

The following source data and figure supplements are available for figure 2:

**Source data 1.** Source data for *Figure 2e,f*.
DOI: https://doi.org/10.7554/eLife.46181.010

**Source data 2.** Source data for *Figure 2g*.
DOI: https://doi.org/10.7554/eLife.46181.011

**Figure supplement 1.** Measurement of the half-life of destabilized GFP in cultured *Drosophila* S2 cells.
DOI: https://doi.org/10.7554/eLife.46181.006

**Figure supplement 1—source data 1.** Source data for *Figure 2—figure supplement 1b*.
DOI: https://doi.org/10.7554/eLife.46181.007

*Figure 2 continued*

**Figure supplement 1—source data 2.** Source data for *Figure 2—figure supplement 1c*.
DOI: https://doi.org/10.7554/eLife.46181.008
**Figure supplement 1—source data 3.** Source data for *Figure 2—figure supplement 1d*.
DOI: https://doi.org/10.7554/eLife.46181.009

strategy makes cloning cumbersome and may render the insertion unstable due to recombination. Thus, we instead decided to use translational enhancing elements that have been demonstrated to increase protein production from mRNA by up to 20 fold (*Pfeiffer et al., 2012*; *Pfeiffer et al., 2010*). These elements include a short 87 bp intervening sequence (IVS) from myosin heavy chain to facilitate mRNA export to the cytoplasm (*Pfeiffer et al., 2010*), a synthetic AT-rich 21 bp sequence (Syn21) to promote translational initiation (*Pfeiffer et al., 2012*; *Suzuki et al., 2006*), and a highly-efficient p10 polyadenylation (polyA) signal from baculovirus (*van Oers et al., 1999*). To test if these elements can be used to increase the reporter signals, we inserted the translational enhancing elements into reporter constructs containing dGFP (destabilized sfGFP) (*Figure 2a–c*). Transgenic flies were generated by phiC31-mediated site-directed integration into the same genomic locus (attP40) to avoid potential position effect (*Groth et al., 2004*). Strikingly, the addition of translational enhancers successfully increased the reporter signal with an expression pattern similar to that of previously reported stable reporters (*Figure 2a–c*) (*Furriols and Bray, 2001*; *Rodrigues et al., 2012*). We further measured the in vivo half-life of the dGFP in live tissues by blocking transcription with Actinomycin (10 µM) and monitored degradation of the GFP (*Figure 2d*). The result shows a reporter half-life ($T_{GFP1/2}$ ~2.6 ± 0.3 hr) similar to what was observed in cultured cells (~3.7 ± 0.7 hr) (*Figure 2d-f*; *Figure 2—figure supplement 1e*). We also tested the absolute signal intensity of regular GFP with the translational enhancer (eGFP) and destabilized GFP with the same enhancer (edGFP). The total signal intensity from edGFP is about 8% of what is observed for eGFP, consistent with a 91% reduction in protein half-life.

## Direct comparison between destabilized and stable reporters in live tissues

The effective increase of FP signal allows us to directly evaluate the activity of short-lived and long-lived traditional reporters. To achieve this, we generated a *STAT::eRFP* reporter with the same translational enhancing elements and inserted it into the same locus (attP40) (*Figure 3—figure supplement 1a*). The activities of *STAT::edGFP* and *STAT::eRFP* reporters (as transheterozygotes) in developing embryos were examined under live imaging conditions (*Figure 3a*). Compared to the stable RFP, the destabilized reporter showed a transient increase in STAT activity in tracheal pits (Tp), pharynx (Pr), proventriculus (Pv), posterior spiracles (Ps), and hindgut (Hg) (*Figure 3a*, *Figure 3—figure supplement 1b*, *Videos 1* and *2*) (*Johansen et al., 2003*). To quantify the temporal changes of dGFP and RFP, the total fluorescent signals of both reporters were measured over time at the indicated region (arrowhead in *Figure 3a*); the dGFP signal shows a definite improvement in response time (*Figure 3b*). Using the half-lives estimated from the in vivo measurement (2.1 hr for dGFP and 18.5 hr for RFP) (*Figure 3—figure supplement 1c–e*) and the reporter synthesis and degradation model (*Figure 1—figure supplement 1*), we further estimate the actual transcriptional activity of the reporter, which happens at an earlier stage (~2 hr in advance of detectable dGFP reporter activity) (*Figure 3b*). This result is consistent with the previously observed temporal expression of the STAT ligand *upd* from stage 9 to 12 (*Johnson et al., 2011*). We also notice that the degradation reaction deviated from first-order kinetics at a high concentration of FP (*Figure 3—figure supplement 1e*). Meanwhile, the degradation speed of the FPs, which depends on the availability of relevant enzymes, may also vary under different conditions. Therefore, the direct interpretation of the difference between dGFP and RFP is the relative rather than absolute ratio. The absolute relationship between the reporters to the actual transcripts should be determined experimentally on a case by case basis.

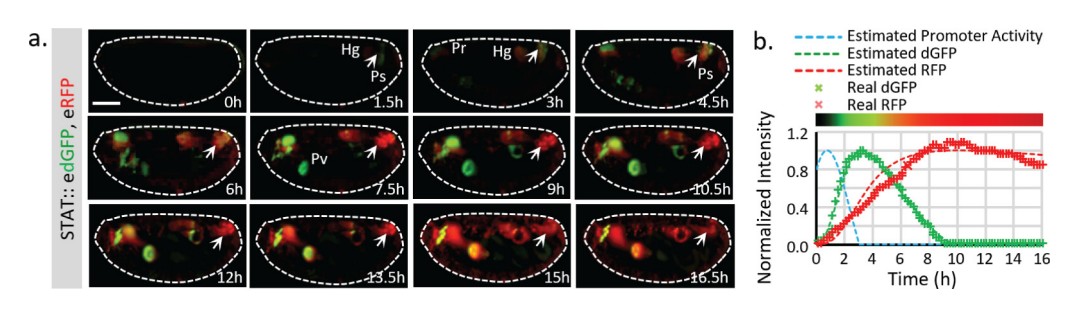

**Figure 3.** Live imaging of both destabilized sfGFP and stable RFP to reveal endogenous STAT dynamics. (a) Time-lapse imaging of developing fly embryos expressing both *STAT::edGFP* and *STAT::eRFP* from stages 12 to 17 when STAT activity starts to increase. Maximum intensity z-projections of the mid-section (120 μm) are shown. Arrowheads indicate the posterior spiracles and hindgut region. The signal from the same structure turns from green to red over time. (b) The total dGFP and RFP fluorescent signals within the posterior spiracles and hindgut region (indicated by arrows in a) are plotted as colored crosses. The estimated promoter activity was calculated using equations (4) (*Figure 1—figure supplement 1b*), and plotted as a dashed blue line. The simulated responses of dGFP and RFP from the estimated transcriptional signal using equations (1)-(3) (*Figure 1—figure supplement 1b*) are plotted as dashed green and red lines respectively. The merged intensity from estimated dGFP and RFP signals is illustrated as a color bar at the top of the plot. Scale bar: 100 μm.

DOI: https://doi.org/10.7554/eLife.46181.012

The following source data and figure supplements are available for figure 3:

**Source data 1.** Source data for *Figure 3b*.
DOI: https://doi.org/10.7554/eLife.46181.016
**Figure supplement 1.** Quantification of STAT reporter dynamics.
DOI: https://doi.org/10.7554/eLife.46181.013
**Figure supplement 1—source data 1.** Source data for *Figure 3—figure supplement 1d*.
DOI: https://doi.org/10.7554/eLife.46181.014
**Figure supplement 1—source data 2.** Source data for *Figure 3—figure supplement 1e*.
DOI: https://doi.org/10.7554/eLife.46181.015

## Combining destabilized GFP and stable RFP to create a transcriptional timer

Short-lived reporters show clear advantages in revealing expression dynamics in live tissues. However, not all tissues are amenable to live imaging. Further, live imaging experiments are time-consuming and hard to adapt for large-scale studies. From our previous analyses, we noticed that dynamic information, including the initiation,

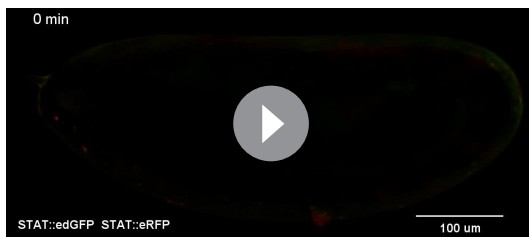

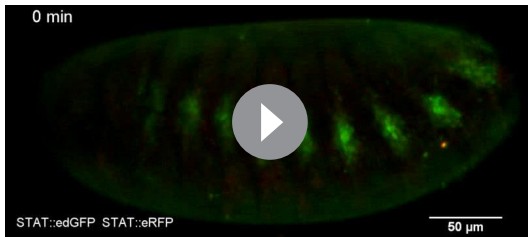

**Video 1.** Live imaging of developing fly embryo (mid projection). Embryo was imaged at room temperature using a Zeiss Lightsheet Z1 microscope with a 20X (N.A. 1.0) lens. Z-stack images (2 μm between each slice) were acquired every 10 min. A maximal intensity z-projection of mid-section (120 μm) is shown in the video. Video was taken from stage 13 to stage 17 when the somatic muscles contract. Genotype of the sample is: w; STAT::edGFP/STAT::eRFP.

DOI: https://doi.org/10.7554/eLife.46181.017

**Video 2.** Live imaging of developing fly embryo (surface projection). Embryo was imaged at room temperature using a Zeiss Lightsheet Z1 microscope with a 20X (N.A. 1.0) lens. Z-stack images (2 μm between each slice) were acquired with 10 min time intervals. A maximal intensity z-projection of surface (25 μm) is shown. Video was taken from stage 13 to stage 17 before the somatic muscles contraction. Genotype of the sample is: w; STAT::edGFP/STAT::eRFP.

DOI: https://doi.org/10.7554/eLife.46181.018

maintenance, and reduction of transcriptional activities, can be directly estimated by comparing the ratio of fluorescence from stable vs destabilized reporters from a still image. Because the GFP matures faster than RFP (maturation time ~0.1 hr for sfGFP [*Khmelinskii et al., 2012*] vs. ~1.5 hr for TagRFP [*Merzlyak et al., 2007*] used in this study), when the promoter switches on, a green signal is detected first. As the promoter remains active, both GFP and RFP signals reach a balance and such that their overlay produces a yellow color (when the max intensities of GFP and RFP are normalized). Furthermore, when transcription switches off, because the GFP signal decreases more quickly (the half-life of dGFP is ~2 hr, and the half-life of RFP is ~20 hr), only the RFP signal is left (*Figure 4a*). Previously, ratiometric imaging of FPs based on their different properties, such as maturation time and sensitivity to specific ions, has been used successfully to measure protein life-time (*Khmelinskii et al., 2012*), pH changes (*Gjetting et al., 2012*; *Zhou et al., 2012*), and $Mg^{2+}$ concentration (*Koldenkova et al., 2015*). We reason that dynamic transcriptional activities might also be detected similarly using the ratio between dGFP and RFP.

To test this strategy in vivo in a multicellular tissue, we examined STAT activity in the larval optic lobe. During development, STAT activity has been shown to act as a negative signal that antagonizes progression of the cell differentiation wave, which triggers the transition of neuroepithelial cells (NEs) into neuroblasts (NBs) (*Yasugi et al., 2008*). A stable STAT reporter shows the spreading of STAT activity within the neuroepithelial region, similar to what we observe with *STAT::eRFP* alone. In contrast, the signal from *STAT::edGFP* together with *STAT::eRFP* revealed a clear 'green front' and 'red rear' in the same region (*Figure 4b*). This dynamic pattern, with a higher STAT activity at the boundary between NEs and NBs, is consistent with previous proposed wave-like STAT activity that propagates from the lateral to medial region (*Yasugi et al., 2008*). In addition, analysis of *STAT::edGFP* and *STAT::eRFP* in fixed larval optic lobes at different developmental stages further support this wave-like propagation model (*Figure 4—figure supplement 1a*).

Another example of the dynamic information revealed by combining edGFP and eRFP is the expression of the Notch reporter in larval brain NBs. NBs undergo an asymmetric cell division to generate smaller progeny (*Figure 4c*). Previous studies have shown that the Notch suppressor complex PON/Numb is preferentially localized to progeny cells (*Suzuki et al., 2006*; *van Oers et al., 1999*) and that ectopic activation of Notch generates a brain tumor phenotype attributable to excess NBs, suggesting that Notch activity is required for NB self-renewal (*Wang et al., 2006*). Strikingly, whereas the stable Notch reporter accumulates in both NBs and their progeny, the destabilized reporter is preferentially expressed in NBs (*Figure 4c*, *Figure 4—figure supplement 1b*), consistent with functional studies (*Wang et al., 2006*). Combining the information from the edGFP and eRFP reporters reveals a clear Notch activity gradient that decreases as NBs differentiate. Importantly, in addition to its superior spatial resolution, the destabilized reporter also shows improved temporal resolution in NBs under live imaging conditions (*Figure 4—figure supplement 2*).

Our data suggest that combining edGFP and RFP creates a useful tool to study transcriptional dynamics, even in fixed samples. To facilitate this, we generated a multicistronic reporter containing both edGFP and RFP connected by the 'self-cleaving' 2A peptide (*Szymczak and Vignali, 2005*), *edGFP-2A-RFP*, which we refer to as a transcriptional timer or 'TransTimer' (*Figure 4—figure supplement 3a*). Larval optic lobes expressing the transcriptional timer controlled by STAT response element show similar expression pattern as transheterozygous *STAT::edGFP* and *STAT::eRFP*, indicating that the multicistronic system is effective (*Figure 4—figure supplement 3b*).

Destabilization of dGFP depends on protein degradation. Potential changes in the degradation speed of dGFP could affect the intensity of dGFP, which might distort our estimation of the real transcriptional activities. To test this possibility, we generated a TransTimer under the control of the fly Ubiquitin promoter (a constitutively active promoter in most fly tissues). *Ubi::edGFP-2A-RFP* shows no significant variation in the green and red ratios in fly embryos or the larval brain (*Figure 4—figure supplement 3c*). In addition, under control of other constitutive promoters, TransTimer also shows a relatively stable ratio between the two colors in different tissues (*Supplementary file 2*), suggesting that changes in the FP ratios observed with TransTimer are primarily due to changes in transcriptional activity in the tested tissues, not cell type or tissue-specific differences in protein degradation. However, for specific organ or developmental stage, a control with a constitutive promoter for protein degradation changes is advisory.

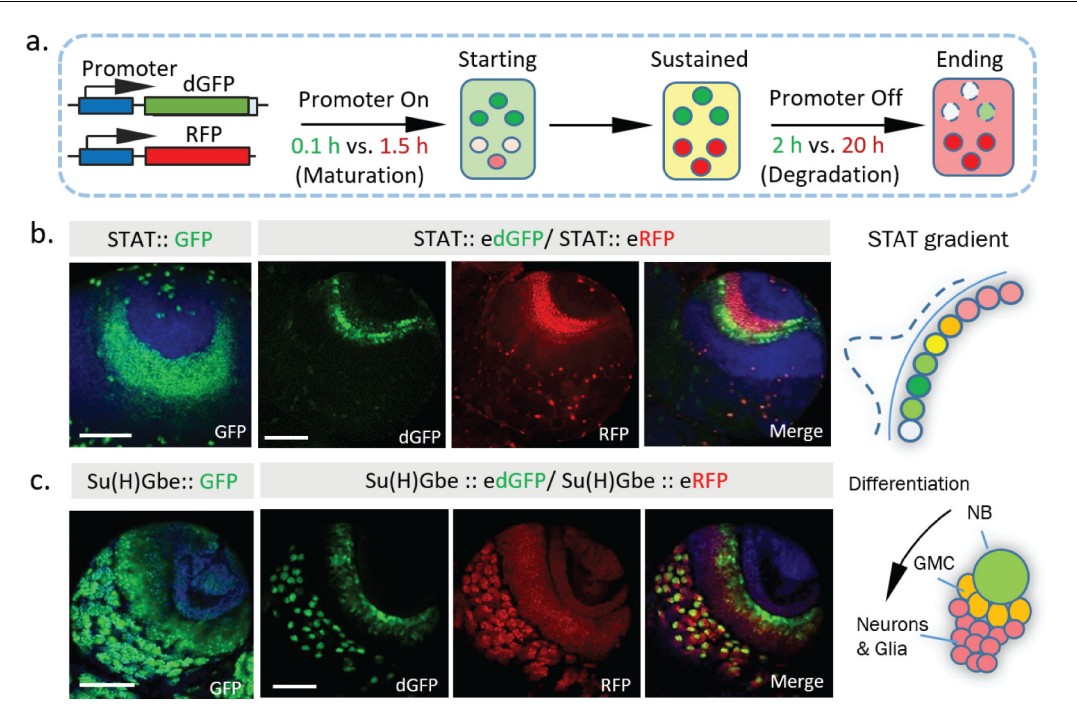

**Figure 4.** Dynamics of STAT and Notch activity revealed in fixed tissues. (a) The destabilized GFP (dGFP) combined with stable RFP functions as a fluorescent timer to reveal transcriptional dynamics. The maturation and degradation half-lives of dGFP and RFP are indicated. (b. c) A comparison between the regular GFP reporter with a combination between dGFP and RFP reporter under control of either STAT response elements or Notch response element Su(H)Gbe. Both reporters are visualized in third instar larval brains. NB, neuroblast; GMC, Ganglion mother cells. Scale bar: (b, c) 50 µm.

DOI: https://doi.org/10.7554/eLife.46181.019

The following source data and figure supplements are available for figure 4:

**Figure supplement 1.** Additional dynamics of Notch and STAT activity revealed in different fixed tissues and stages.
DOI: https://doi.org/10.7554/eLife.46181.020

**Figure supplement 2.** Live imaging of Notch activity in dividing Type I neuroblasts (NBs) in larval brain.
DOI: https://doi.org/10.7554/eLife.46181.021

**Figure supplement 2—source data 1.** Source data for *Figure 4—figure supplement 2b*.
DOI: https://doi.org/10.7554/eLife.46181.022

**Figure supplement 3.** Combination of edGFP and RFP into a single construct using 2A peptide.
DOI: https://doi.org/10.7554/eLife.46181.023

## Generation of a transcriptional timer for use with the UAS/Gal4 system

Creating a TransTimer reporter for new signaling or target genes requires cloning of different signal response elements. Meanwhile, several large transgenic collections (up to several thousands) of Gal4 lines under the control of enhancers of different genes have been generated (*Brand and Perrimon, 1993*; *Jenett et al., 2012*; *Lee et al., 2018*). A TransTimer controlled by UAS would provide a quick way to test expression dynamics of Gal4 lines using a simple genetic cross (*Figure 5a*). Thus, we generated *UAS::TransTimer* transgenic flies and tested the UAS-controlled version in the adult *Drosophila* gut. TransTimer under control of *esg-Gal4*, a fly intestinal stem cells (ISCs) and enteroblasts (EBs) driver (*Micchelli and Perrimon, 2006*; *Ohlstein and Spradling, 2007*), revealed particular cells within the stem cell group that turned 'red,' distinguishing them from other 'yellow' stem cells (*Figure 5b*). Further analysis of these 'red' cells showed that they down-regulate *esg* expression and up-regulate the enteroendocrine cell marker Prospero (Pros), suggesting that they are differentiating (*Figure 5—figure supplement 1a*). In addition, the *esg*-controlled TransTimer shows significantly more dynamics in the developing intestine and regenerating intestine following Bleomycin treatment

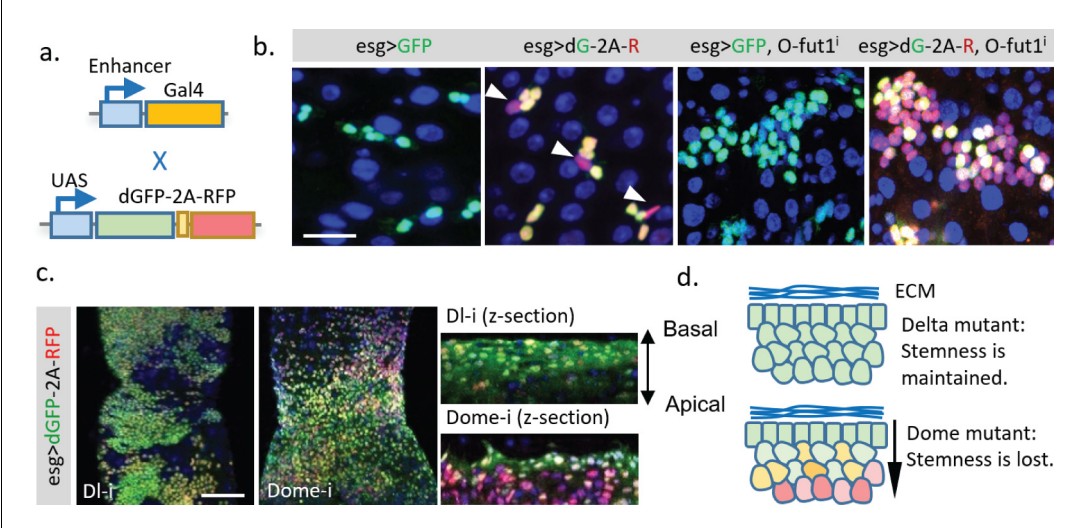

**Figure 5.** Application of the dGFP-P2A-RFP dynamic reporter to study gene expression dynamics of fly intestine stem cells. (**a**) *UAS-TransTimer*, a UAS controlled multicistronic reporter containing dGFP and RFP connected by the P2A peptide was generated and crossed with the intestine stem cell driver *esg-Gal4*. (**b**) Compared with the regular GFP reporter, *TransTimer* (*dGFP-P2A-RFP/dG-2A-R*) reveals the differentiating cells, which reduce the expression of the *escargot* (*esg*) stem cell marker (arrowheads). Knocking-down *O-fut1* inhibits Notch activity and causes ISC tumors. Compared to regular GFP, significant heterogeneity is revealed in the over-proliferating cell clusters using TransTimer. (**c,d**) Intestine tumors are generated by knocking down the Notch ligand *Dl* or the cytokine receptor *Domeless* (*Dome*). The double-headed arrow indicates the z-direction with the basal side of intestine epithelium facing up and apical side facing down. Scale Bar: (**b**) 25 µm; (**c**) 100 µm.

DOI: https://doi.org/10.7554/eLife.46181.024

The following figure supplement is available for figure 5:

**Figure supplement 1.** Dynamic activity of TransTimer in the fly intestine.
DOI: https://doi.org/10.7554/eLife.46181.025

(a DNA damaging agent) (*Amcheslavsky et al., 2009*), consistent with the different level of activity of stem cells under these conditions (*Figure 5b*, *Figure 5—figure supplement 1b*).

Next, we used TransTimer to examine the cell heterogeneity of different intestine tumors. In ISC tumors induced by knocking down *O-fut1*, an enzyme required for Notch maturation (*Micchelli and Perrimon, 2006*; *Ohlstein and Spradling, 2007*), the regular GFP reporter showed only a moderate variation of the stem cell marker *esg*. By contrast, TransTimer revealed an evident decrease of dGFP compared to RFP in ~70–60% of the cells in the cluster, suggesting that a substantial heterogeneity in the tumor is caused by down-regulation of *esg* over time (*Meacham and Morrison, 2013*) (*Figure 5b*). Tumors generated by knocking down either *Delta*, the ligand of Notch receptor, or *Domeless* (*Dome*), the transmembrane receptor of JAK/STAT signaling pathway, grow into a similar multilayered cell cluster (*Figure 5c*) (*Jiang et al., 2009*). Interestingly, compared to *Dl* mutant tumors, where all multilayered cells maintain constant levels of *esg* (dGFP/RFP ratio), the inner layer of *Dome* mutant tumors shows clear reduction of *esg* expression (lower dGFP/RFP ratio) relative to the basal layer. This result suggests that *Dome* mutant tumors, unlike *Dl* mutant tumors, require direct contact with the basal membrane to keep their stemness (*Figure 5d*).

## Application of TransTimer for the discovery of new genes with dynamic expression

As demonstrated above, *UAS-TransTimer* is an effective tool to discover expression changes when crossed with a Gal4 driver of interest. To further test the power of this approach to discover new genes with interesting expression dynamics, we screened ~450 Gal4 lines using *UAS-TransTimer* (*Marianes and Spradling, 2013*). 37 lines (~8%) showed clear dynamic activities (substantial variation in dGFP/RFP ratio) in either larval brain, imaginal disc, or adult intestine (*Figure 6a*, *Supplementary file 1*), whereas the remaining Gal4s showed essentially uniform dGFP/RFP ratios,

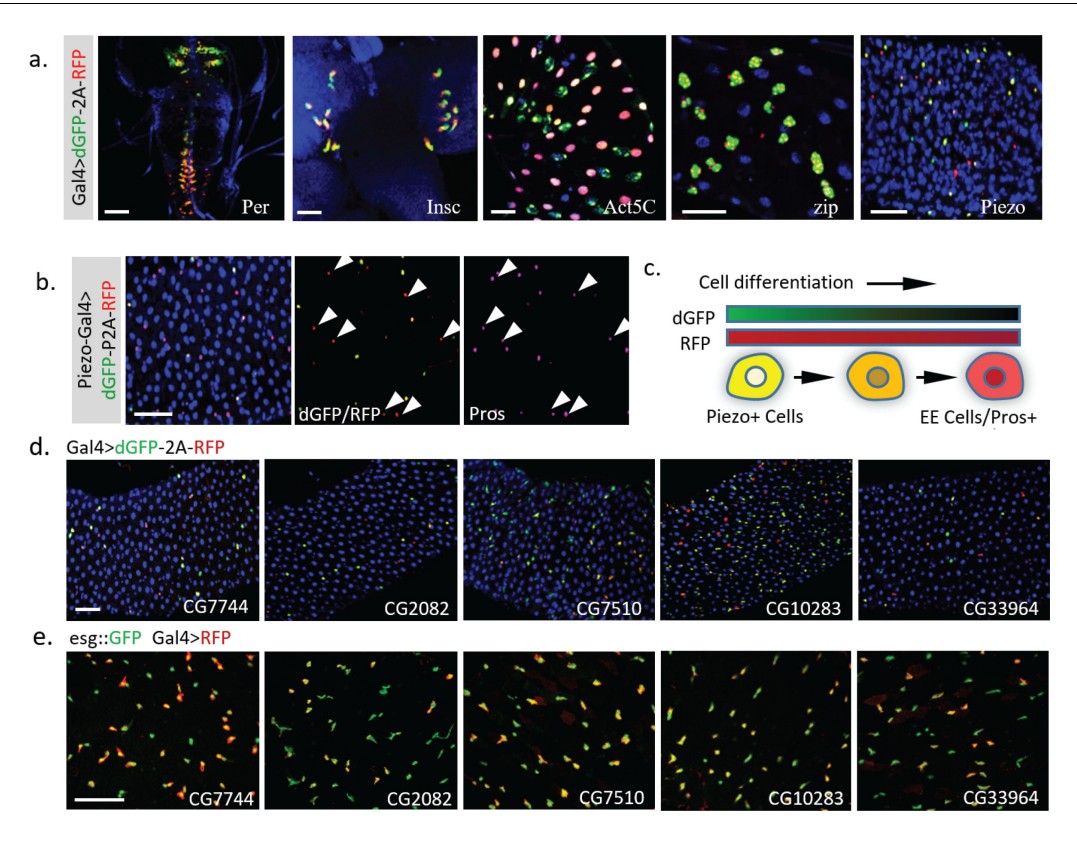

**Figure 6.** Dynamic pattern of TransTimer driven by different Gal4 drivers. (a) Examples of Gal4 lines that show clear dynamic patterns with TransTimer (*UAS-dGFP-P2A-RFP*) in various organs: *Per-Gal4*, controlled by the circadian rhythm regulator Period (larval brain and ventral ganglion); *insc-Gal4*, expressed in Type II neuroblasts (larval brain); *Act5C-Gal4*, controlled by the *act5C* enhancer (larval intestine); *GMR51F08-Gal4*, controlled by the enhancer from the fly myosin heavy chain *Zipper* (larval intestine, adult intestine precursor cells); *Piezo-Gal4 (BL58771)*, controlled by the cloned enhancer from mechanical sensitive ion channel *Piezo* (adult intestine). (b) Dynamic reporter under control of *Piezo-Gal4[KI]*, a *Gal4* knock-in after the first ATG of *Piezo*. Piezo + cells with high RFP and low GFP are positive for the EE cell marker Prospero (Pros). (c) Illustration of the differentiation process from Piezo + EP (enteroendocrine precursor) to Pros + EE cells. (d) The expression dynamics of *UAS-TransTimer* driven by different Gal4 lines in the fly intestine. (e) Gal4 activities are detected in a subpopulation of the *esg* +stem cells. Stem cells are marked by *esg::GFP* and the expression of Gal4s are revealed by *UAS-RFP*. Scale bar: (a-e) 50 μm.
DOI: https://doi.org/10.7554/eLife.46181.026

The following figure supplement is available for figure 6:

**Figure supplement 1.** Effect of using Gal4/UAS system on the transcriptional timer.
DOI: https://doi.org/10.7554/eLife.46181.027

suggesting stable expression (images of representative control Gal4 lines are shown in *Supplementary file 2*). Among the genes with dynamic expression patterns, we discovered the mechanosensitive channel *Piezo*, which is expressed in the posterior midgut specifically in EE precursor cells (*He et al., 2018*). *TransTimer* driven by *Piezo-Gal4* displays a spatially dynamic expression pattern (separation between the 'green' and 'red' signals) (*Figure 6b*). In addition, the 'red' cells, which down-regulate Piezo expression, are positive for the EE cell marker Pros, consistent with the results of our previous study showing that Piezo + cells differentiate into EE cells (*Figure 6c*). In addition to *Piezo*, we also identified new uncharacterized genes with dynamic expression patterns in a subpopulation of *esg+* cells (*Figure 6d*, *Figure 6e*). Further studies of these genes will be required to determine whether they are markers of partially differentiated cells like Piezo or if their expression levels oscillate in the stem cells.

Although *UAS-TransTimer* is a useful tool to explore the hidden transcriptional dynamics using the Gal4 collection, the addition of Gal4 as an intermediate also alters the signal property of Trans-Timer. According to our model, the presence of Gal4 has two major effects: first, as an amplification

mechanism, it can enhance the output signal, which may be advantageous for some weak enhancer (*Figure 6—figure supplement 1a,b,f*); second, it delays the dynamic of the reporter, as the new half-life of TransTimer controlled by Gal4 is generally proportional to the sum of $T_{G1/2}$ (half-life of Gal4) and $T_{FP1/2}$ (half-life of FPs) (*Figure 6—figure supplement 1c–e*). Because the exact half-life of Gal4 in vivo is unknown, we estimated the effect of *Gal4/UAS* system by comparing the dGFP and RFP controlled directly by the Notch responding element *Su(H)Gbe* and *UAS-TransTimer* controlled by *Su(H)Gbe-Gal4*. Consistent with the predictions of the model, the *UAS-TransTimer* that is driven by *Su(H)Gbe-Gal4* shows a similar but considerable broader Notch activation pattern in the third instar larval wing discs than *Su(H)Gbe-TransTimer*, which is probably due to both longer signal retention and stronger signal amplification (*Figure 6—figure supplement 1g*). In the larval brain, the *UAS-TransTimer* shows a similar activation gradient in neuroblast cells and their progenies but with significantly stronger retention in daughter cells due to the slow signal reduction (*Figure 6—figure supplement 1h*).

To overcome the delay effect of Gal4, we decided to control *TransTimer* directly by endogenous promoters. To achieve this goal, we generated transgenic flies with *TransTimer* controlled by a minimal synthetic *Drosophila* promoter that is silent unless activated by nearby enhancers (*Pfeiffer et al., 2008*), and randomly mobilized the transgene in the fly genome to identify endogenous enhancers with dynamic activities (*Figure 7a*, *Figure 7—figure supplement 1*). After screening ~400 independent enhancer trap lines, we identified 46 unique lines that showed fluorescent signals in the larval brain, imaginal disc, or adult intestine. 17 of these 46 lines show clear expression dynamics, suggesting that TransTimer can detect expression changes at endogenous levels (*Figure 7b*, *Figure 7—figure supplement 1b*, *Supplementary file 3*). To validate the screen, we tested the expression and function of new genes identified in this enhancer trap screen. Since we are particularly interested in new lines that show exclusive expression in stem cells, we chose TransTimer insertions near the promoters of *Tsp42Ea*, a Tetraspanin protein, and *CG30159*, an evolutionarily conserved gene with unknown function - as the function of these genes had yet not been characterized in fly intestine. A Gal4 line (*NP1176-Gal4*), located closely (within 250 bp) to the TransTimer insertion site at the promoter of *Tsp42Ea* and *CG30159*, also shows specific expression in both larval and adult intestine stem cells (*Figure 7d*), which is very similar with the expression pattern revealed by TransTimer (*Figure 7b*). Knocking down *CG30159* significantly reduces stem cell numbers, suggesting that *CG30159* is required for maintenance of intestinal stem cell (*Figure 7e,f*). The human homolog of *CG30159* is *C3orf33*, which has been identified as a regulator of the extracellular signal-regulated kinase (ERK) and predicted to be a secreted peptide due to the presence of signal peptide at its N-terminus (*Hao et al., 2011*). Its function in intestinal stem cells requires further investigation.

As we have shown above, the enhancer trap screen with TransTimer can effectively detect expression dynamics in vivo. However, this screen can only detect gene expression and cannot be used to manipulate the target cell population. To extend the application of TransTimer, we replaced the RFP with lexA, a yeast transcriptional factor used as a binary expression system together with its binding sequence lexA operator (lexAop) (*Yagi et al., 2010*). This *dGFP-P2A-lexA* construct can not only detect expression dynamics when crossed with lexAop-RFP but also manipulate gene expression in labeled cells in the presence of an additional lexAOP-controlled transgene (*Figure 7g*). We refer to *dGFP-P2A-lexA* as 'TransTimerLex'. To test the feasibility of this strategy, we generated transgenic flies containing the TransTimerLex insertion and randomly mobilized the element in the fly genome. From our pilot screen (~20 independent lines), we identified one insertion under control of Larp, a transcriptional factor, which shows clear expression dynamics in the larval intestine (*Figure 7h*). This result suggests that the TransTimeLex system will be a useful way to both identify new genes and manipulate gene expression in the corresponding cells.

## Discussion

In this study, we described a general and straightforward strategy to use destabilized transcriptional reporters in vivo and demonstrated its power in revealing the spatio-temporal dynamics of gene expression, which is missed by conventional transcriptional reporters. In addition, we generated a dual-color TransTimer that encodes the transcriptional dynamics into a green-to-red color ratio which can be analyzed in fixed tissues. This TransTimer provides a unique opportunity for large-scale

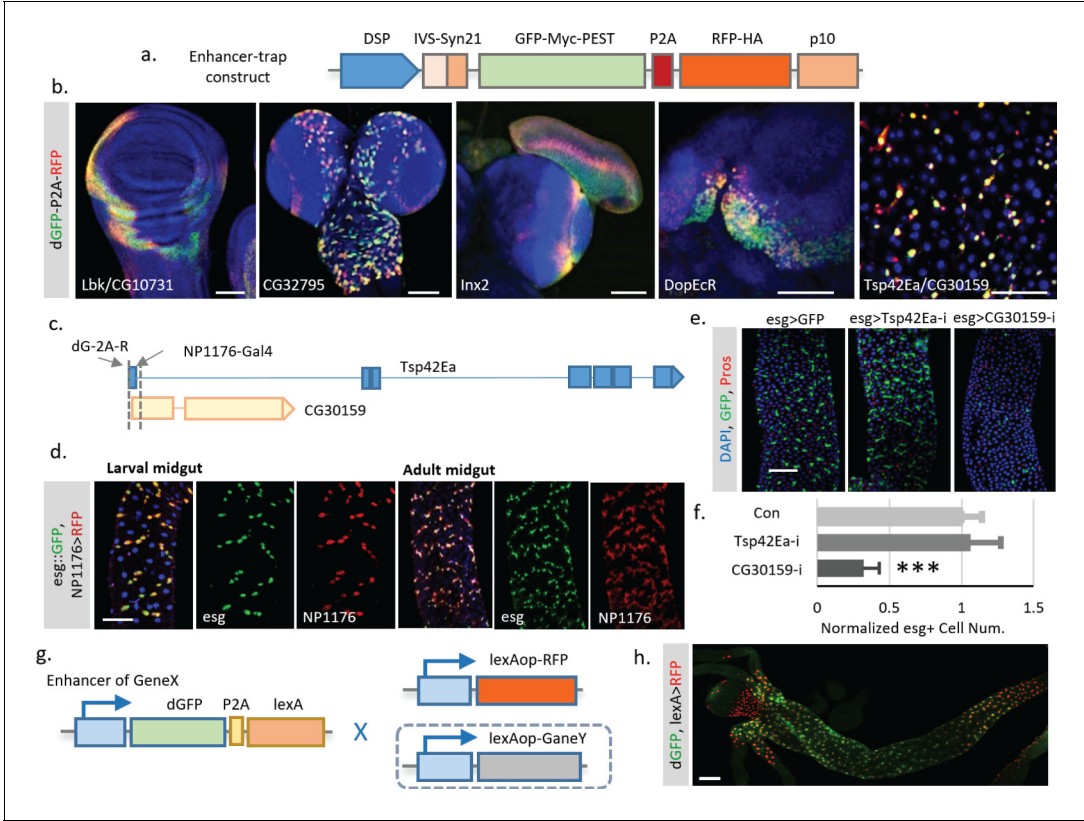

**Figure 7.** Enhancer trap screen for endogenous transcription dynamics. (**a**) A P-element containing *Drosophila* synthetic core promoter (DSCP), translational enhancing elements, and *dGFP-P2A-RFP* (TransTimer) was randomly mobilized in the fly genome to identify enhancers with dynamic expression. edGFP and RFP are tagged by Myc and HA epitopes, respectively, to allow signal enhancement by immunohistochemistry. (**b**) Examples of enhancer trap lines that show dynamic patterns in different organs: *Lbk,* an Immunoglobulin-like protein, or *CG10731*, the subunit S of mitochondrial ATP synthase complex (wing disc); *CG32795*, a novel membrane protein (larval brain); *inx2*, a gap junction protein (larval brain and eye disc); *DopEcR*, Dopamine and Ecdysteroid receptor (larval brain); and *Tsp42E*, a tetraspanin protein; and *CG30159,* a novel gene with unknown function (adult intestine). (**c**) The *dGFP-P2A-RFP* and the *NP1176-Gal4* enhancer trap insertions at the promoter region of *Tsp42Ea* and *CG30159*. Exons are shown in the diagram (blue for *Tsp42Ea*, yellow for *CG30159*). *Tsp42Ea* and *CG30159* share the same promoter region. (**d**) *NP1176-Gal4* (marked by *UAS-RFP*) shows stem cell expression like *esg* (marked by *esg::GFP*). (**e, f**) Knocking down *CG30159* significantly reduces stem cell numbers (*esg+* cells) in the intestine, while RNAi against *Tsp42Ea* does not have a significant phenotype. Number of *esg* +cells were quantified within 100 μm̂two regions from n = 8 (*GFP* control), n = 7 (*Tsp42Ea-i*), and n = 9 (*GC30159-i*) adult fly intestines. p-value<0.001. Cell numbers are normalized according to the control. (**g**) Schematic of the TransTimerLex for enhancer trap with RFP replaced by the bacterial transcriptional repressor LexA. This reporter allows both two-color contrast and further genetic manipulation of the target cell population using the LexA/lexAop binary expression system. (**h**) An enhancer trap line, under potential control of *Larp* enhancer, was crossed with *lexAop::RFP*. The anterior region of the larval intestine is shown, revealing a decrease of transcriptional activity in the proventriculus. Scale bar: Scale bar: (**b, h**) 50 μm; (**d, e**) 100 μm.

DOI: https://doi.org/10.7554/eLife.46181.028

The following source data and figure supplement are available for figure 7:

**Source data 1.** Source data for *Figure 7f*.
DOI: https://doi.org/10.7554/eLife.46181.030
**Figure supplement 1.** Enhancer trap screen for gene expression dynamics.
DOI: https://doi.org/10.7554/eLife.46181.029

screens for in vivo expression dynamics in all types of tissues. Further, our study indicates that Trans-Timer is effective for the discovery of new genes with interesting expression patterns, either using a candidate gene approach or random genome-wide screening. Our reporter system may also be combined with other techniques such as FACS-seq and signal cell sequencing techniques to provide a time-dependent change of the transcriptome in vivo. In fact, a similar strategy has recently been successfully applied to provide the temporal information for signal cell sequencing of the mouse

intestinal stem cell system (*Gehart et al., 2019*). Therefore, we expect that this new method will widely facilitate studies in *Drosophila* and other organisms.

## Materials and methods

Molecular Biology cDNAs of sfGFP and TagRFP, codon optimized for *Drosophila*, were a kind gift from Dr. Hugo Bellen (*Venken et al., 2011*). HSPmini-IVS-Syn21 and p10 were amplified by PCR from pJFRC81 (Addgene) (*Pfeiffer et al., 2012*). MODC sequence was from pBPhsFlp2::PEST vector (Addgene) (*Nern et al., 2011*). The vector containing LHG, which encodes a chimeric protein consisting of the LexA DNA binding domain and the Gal4 transcriptional activation domain, was a kind gift from Dr. Konrad Basler (*Yagi et al., 2010*). pUAST4 and pWALIUM10-roe were from our lab stock collection. Primers and gBlocks were obtained from Integrated DNA Technologies. PCR was performed with the proofreading enzyme Phusion (NEB). Plasmid purification, PCR purification, and gel extraction were performed with QIAprep Spin Miniprep Kit (QIAGEN), QIAquick PCR Purification, and QIAquick Gel Extraction Kits (QIAGEN), respectively. In-Fusion cloning and Gateway cloning were performed using In-Fusion HD Liquid Kits (Clontech), and BP and LR Clonase Enzyme Mixes (ThermoFisher Scientific). All cloning experiments were verified by DNA sequencing.

The full names of plasmids (1-24) generated in this study are listed in *Supplementary file 4*. sfGFP, nls-sfGFP, or nls-TagRFP were first cloned into the pDONR221 vector by gateway BP cloning. The MODC sequence was inserted before the stop codon of pENTR-sfGFP and pENTR-nls-sfGFP using In-Fusion. Mcd8 and His2A were amplified by PCR and inserted before the ATG of pENTR-sfGFP-MODC linearized by PCR by the In-Fusion assembly. 3XMyc and 3XHA were inserted after GFP and RFP by In-Fusion. Plasmids 1–6 for cell culture were generated by recombining the pENTR constructs into pUbi-Gateway vector (pUWR). To generate the empty reporters (plasmids 7 to 9), pCasper4B2G was first opened by SpeI. PCR fragments of HSPmini-IVS-syn21 and nls-sfGFP (or nls-TagRFP) were assembled into the vector with the first SpeI cutting site mutated. Then, the vector was opened again by SpeI, and PCR fragment of p10 polyA was inserted using In-Fusion to yield pCaSpeR4B2G-IVS-syn21-nlsGFP-p10. MODC (PEST) was amplified by PCR and added into the SpeI site by In-Fusion. Plasmid 10 was constructed similarly with HSPmini, nlsGFP-PEST, and SV40 poly(A) sequences inserted into the SpeI site of pCaSpeR4B2G. Plasmids 11–18 were generated by insertion of PCR products of 6XSTAT or Su(H)Gbe into the XbaI site of the corresponding empty reporter plasmids. PCR fragments of HSPmini-IVS-syn21-nlsGFP-MODC and 2A-nlsRFP-p10 were assembled into the pENTR vector linearized by PCR. Then the HSPmini-IVS-syn21-nlsGFP-MODC-2A-nlsRFP-p10 was PCR amplified and replaced the UAS and Gateway cassette of the pWALIUM10-roe to generate plasmid 19. 6XSTAT was amplified by PCR and inserted at the before the HSPmini of 19 to create plasmid 20. The Ubi promotor was amplified from pUWR vector to replace the UAS-HSPmini promoter of plasmid 21. The IVS-syn21-nlsGFP-MODC-2A-nlsRFP-p10 was PCR amplified and inserted into the pWALIUM10-roe by replacing the Gateway cassette to generate plasmid 22. Then, 3XMyc and 3XHA were inserted after GFP and RFP of pENTR-HSPmini-IVS-syn21-nlsGFP-MODC-2A-nlsRFP-p10, respectively. The HSPmini promoter was replaced by *Drosophila* synthetic core promoter (DSCP), which contains multiple core promoter motifs, to increase the chance of trapping endogenous enhancers. The pENTR-DSCP-IVS-syn21-nls-sfGFP-3XMyc-MODC-2A-nlsRFP-3XHA-p10 was recombined into pCaSpeR4-Gateway vector using LR cloning (plasmid 23). LexA was amplified by PCR and replaced the nlsRFP-3XHA of pENTR-DSCP-IVS-syn21-nls-sfGFP-3XMyc-MODC-2A-nlsRFP-3XHA-p10. Then pENTR-DSCP-IVS-syn21-nls-sfGFP-3XMyc-MODC-2A-LexA-p10 was recombined into pCaSpeR4-Gateway vector using LR cloning (plasmid 24).

### Drosophila genetics

The following fly lines were obtained from the Bloomington Drosophila Stock Center: Delta2-3(99B) (BL3629), Dorothy-Gal4 (BL6903), Dome-RNAi (BL32860), CG30159-*RNAi (BL61888), Tsp42Ea-RNAi (BL39044), fz-Gal4 (BL66817), Per-Gal4 (BL7127), Trx-Gal4 (BL40367), ZnT41F-Gal4 (BL66859), Ogre-Gal4 (BL49340), Rho-Gal4 (BL45254), Gcm-gal4 (BL35541), igl-Gal4 (BL76744), dMyc-Gal4 (BL47844), Hh-Gal4 (BL49437), Antp-Gal4 (BL26817), Plc21C-Gal4 (BL76142), anchor-Gal4 (BL66861), tutl-Gal4 (BL66824), Act5C-Gal4 (BL4414), zip-Gal4 (BL48187), dMyc-Gal4 (BL47844), piezo-Gal4 (BL58771, BL59266), ppk-Gal4 (BL32078, BL32079), Ubi-Gal4 (BL32551), MESK2-Gal4 (BL67434), fz-Gal4 (BL66817), Mip-Gal4 (BL51984), CG7744-Gal4 (BL76662), CG4467-Gal4 (BL66843), CG2082-Gal4*

(BL76181), CG7510-Gal4 (BL66861), CG10283-Gal4 (BL76152), CG33964-Gal4 (BL76742), CG14995-Gal4 (BL76721), CG5521-Gal4 (BL76180), CG8177-Gal4 (BL77781), CG34347 (BL76674), CG13175-Gal4 (BL76742), CG8270-Gal4 (BL77741), CG15270-Gal4 (BL76649), CG43980-Gal4 (BL66863), CG40006-Gal4 (BL), CG43980-Gal4 (BL66863), and dMef2-Gal4 (BL27390). NP1176-Gal4 was from DGRC (Kyoto Stock Center). Dl-RNAi (v37287) was from Vienna Drosophila Resource Center. GMR-Gal4, Da-Gal4, tubGal4, and esg-Gal4 were from lab stocks. Insc-Gal4, ase-Gal80ts was a gift from Dr. Dong Yan (Zhu et al., 2011), and Dl-Gal4 and Su(H)Gbe-Gal4 were from Dr. Steve Hou (Amcheslavsky et al., 2014). All flies were maintained on cornmeal-yeast-agar media. Stocks were kept at room temperature with a 12/12 light/dark cycle.

## Drosophila S2R + cell culture and Western blotting

Drosophila S2R + cells were grown in Schneider's Drosophila Medium (SDM) (Invitrogen) containing 10% heat-inactivated fetal bovine serum (FBS) at 25°C. Sub-confluent S2R + cells were seeded in 6-well plates and subsequently transfected using Effectene Transfection Reagent (QIAGEN). Cells were cultured for 48 hr before experiments. 10 µM (final concentration) Actinomycin D was used to block RNA synthesis, and 100 µg/ml (final concentration) cycloheximide was used to block protein synthesis. Cells were treated with the indicated drugs up to 4.5 hr before significant cell death was observed. Plasmids expressing pUbi-dGFP-Myc (0.03 ug), and pUbi-RFP-HA (0.01 ug), together with empty plasmids (to reach a total of 0.3 ug of DNA) were added in each 6-well plate during transfection. The dilution of the expression plasmid was important: we observed that too much protein expression saturates the degradation machinery and prolongs the observed half-life. S2R + cells were harvested by centrifugation and lysed in RIPA buffer. Proteins were separated on a 10% SDS-PAGE gel and analyzed by Western blotting. Quantitative Western blots were performed as previously described (Eaton et al., 2014). Images were acquired using a LI-COR Odyssey Classic imager and analyzed using NIH ImageJ.

## Generation and test of enhancer trap lines using dynamic enhancers

Dynamic reporters were integrated into the fly genome using P-element mediated transformation by injection into w1118 embryos (Rubin and Spradling, 1982). Transgenic lines were balanced and mapped using w*; Sco/CyO; MKRS/TM6B. Then, six independent lines were generated by crossing with w*;Sp/CyO; Sb,P(Delta2-3)99B/TM6B,Tb+ (BL3629). Males from the F1 generation with red eyes and carrying the CyO balancer were further crossed individually with w1118 females. F2 males with red eyes that co-segregate with the CyO balancer were used in the initial screen.

Detailed crosses for the enhancer screen are shown in Figure 7—figure supplement 1a. ~400 fly lines were recovered from the F2 generation. Third instar larval brains, imaginal discs, and adult intestines of these flies were dissected and examined for GFP and RFP signal using a Zeiss LSM 780 confocal microscope.

P-element insertion sites were mapped by Splinkerette PCR (Horn et al., 2007). PCR primers specific for 5 and 3 prime ends of P-elements were used as previously described (Potter and Luo, 2010). Genomic sequences flanking the P-element insertion sites were recovered and shown in Supplementary file 5. These sequences were used in BLAST searches against the Drosophila Genome Database.

## Immunostaining

Immunostainings of Drosophila intestines were performed as previously described (Micchelli and Perrimon, 2006). The following antibodies were used: mouse anti-Prospero (1:50, Developmental Studies Hybridoma Bank), mouse anti-HA (1:500, Abcam, ab18181), rabbit anti-Myc tag (1/250, Cell signaling), goat anti-mouse IgGs conjugated to Alexa 647 (1:500, Molecular Probes), mouse IgGs conjugated to Alexa 488 (1:500, Molecular Probes), IRDye 800CW Goat anti-Rabbit IgG (1:10,000 LI-COR P/N 926–32211), and IRDye 680RD Goat anti-Mouse IgG (1:20,000 LI-COR P/N 926–68070). Dissected fly tissues were mounted in Vectashield with DAPI (Vector Laboratories). In all micrographs, the blue signal shows the nuclear marker DAPI. Fluorescence micrographs were acquired with a Zeiss LSM 780 confocal microscope. All images were adjusted and assembled in NIH ImageJ.

## Modeling of FP production, maturation, and degradation

The model used to calculate reporter synthesis, maturation, and degradation was modified from previously described equations (*Wang et al., 2008*) with the addition of an equation for mRNA degradation. Briefly, degradation of mRNA and the synthesis rate of premature (nonfluorescent) protein (NP) is proportional to the mRNA concentration (R). Generation of the mature reporter (MP), modeled as a first-order chemical reaction, only depends on the concentration of NP. Protein degradation is modeled independently of the maturation process. The degradation rates of mRNA and proteins are first modeled based on Michaelis-Menten (MM) function (*Figure 1*-figure supplement b, equations 1-3), which considers the potential saturation of the degradation machinery. When the substrate concentration is significantly smaller than the Michaelis constant $Km$, the equations can be simplified with the half-life of the mRNA and protein explicitly displayed (*Figure 1—figure supplement 1b*, equations 1'−3'). Dilution by cell division is not included in this model because the fluorescent signal is analyzed in a cell cluster rather than in individual cells, and cell division does not affect the total intensity from the entire cell group and no significant changes in degradation speed have been observed between different cells (*Figure 3—figure supplement 1*). With this first-order kinetic model, the transcriptional activity of the promoter $F(x)$ can be calculated through equation from the observed fluorescent reporter signal $[MP]$ (4). For sfGFP, the maturation time is ~0.1 hr, which is much smaller than its protein half-life, such that equation 4 can be further simplified as 4'. To calculate $F(x)$, the dGFP signal was fitted with a polynomial function (order = 4) to generate the first and second derivatives.

## Live-cell imaging and data analysis

Live-cell imaging of developing embryos and dissected larval brains was performed as previously described (*Tomer et al., 2012*; *Lerit et al., 2014*; *Lemon et al., 2015*). Images were captured on a Zeiss Lightsheet Z1 microscope using a 20X (N.A. 1.0) lens. A z-stack of the dual-color image (488 nm excitation/500–550 nm detection for GFP, and 561 nm excitation/580–650 nm detection for RFP) was recorded at 10 min intervals. This interval was chosen empirically to minimize photobleaching without losing temporal information. Photobleaching was measured by continuously imaging of the sample for 50 frames for 10 min and adjusted during image processing. Images of fixed tissue were captured on a Zeiss LSM 780 confocal microscope. Total fluorescent intensity in 3D volume was acquired using Imaris image analysis software (Bitplane). The rest of the analysis was completed using NIH ImageJ with customized macros. Simulation of the model was completed in MATLAB. The Student's unpaired, two-tailed t-test was used to determine statistical significance between samples.

## Acknowledgements

We thank Douglas Richardson at Harvard Biological Imaging Center for technical support and advice, and Ben Ewen-Campen, Benjamin Housden, Stephanie Mohr, and Muhammad Ahmad for comments on the manuscript. This work was supported by the Damon Runyon Cancer Research Foundation (LH) and NIH (R21DA039582). NP is an investigator of the Howard Hughes Medical Institute.

## Additional information

### Funding

| Funder | Grant reference number | Author |
| --- | --- | --- |
| Damon Runyon Cancer Research Foundation | | Li He |
| National Institute of General Medical Sciences | R21DA039582 | Norbert Perrimon |
| Howard Hughes Medical Institute | | Norbert Perrimon |

The funders had no role in study design, data collection and interpretation, or the decision to submit the work for publication.

## Author contributions

Li He, Conceptualization, Data curation, Formal analysis, Supervision, Funding acquisition, Validation, Investigation, Methodology, Writing—original draft; Richard Binari, Data curation, Methodology, Performed the enhancer screen for dynamic expression patterns in over 500 fly lines; Jiuhong Huang, Data curation, Formal analysis, Performed the construction, in vitro, and in vivo characterization of destabilized fluorescent reporters; Julia Falo-Sanjuan, Data curation, Performed the in vivo characterization of destabilized fluorescent reporters in different fly tissues; Norbert Perrimon, Supervision, Funding acquisition, Project administration, Writing—review and editing

## Author ORCIDs

Li He (ID) https://orcid.org/0000-0003-2155-606X
Norbert Perrimon (ID) https://orcid.org/0000-0001-7542-472X

## Decision letter and Author response

Decision letter https://doi.org/10.7554/eLife.46181.039
Author response https://doi.org/10.7554/eLife.46181.040

# Additional files

### Supplementary files

• Supplementary file 1. Gal4s with dynamic expression patterns. (Gal4s analyzed in the main figures were not listed.)
DOI: https://doi.org/10.7554/eLife.46181.031

• Supplementary file 2. Gal4s without significant expression dynamics in different fly organs.
DOI: https://doi.org/10.7554/eLife.46181.032

• Supplementary file 3. Enhancer trap lines of Transcriptional Timer (HSPmini-IVS-p21-nlsGFP-Myc-PEST-P2A-nlsRFP-HA-p10) with expression dynamics in different fly organs.
DOI: https://doi.org/10.7554/eLife.46181.033

• Supplementary file 4. Cloning information and sequences of constructs.
DOI: https://doi.org/10.7554/eLife.46181.034

• Supplementary file 5. Mapping Results of enhancer trap lines by splinkerette PCR.
DOI: https://doi.org/10.7554/eLife.46181.035

• Supplementary file 6. Key resources table.
DOI: https://doi.org/10.7554/eLife.46181.036

• Transparent reporting form
DOI: https://doi.org/10.7554/eLife.46181.037

### Data availability

All essential data are provided in the supplementary materials. All the reagents created by this study (plasmids and transgenic flies) will be donated to public domains including Addgene and Bloomington Stock Center.

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
