## [Decision Letter]

Thank you for submitting your article "In vivo study of gene expression with an enhanced dual-color fluorescent transcriptional timer" for consideration by *eLife*. Your article has been reviewed by three peer reviewers, and the evaluation has been overseen by K VijayRaghavan as the Senior Editor and Hugo Bellen as the Reviewing Editor. The following individual involved in the review of your submission has agreed to reveal their identity: Julie Simpson (Reviewer #3).

The reviewers have discussed the reviews with one another and the Reviewing Editor has drafted this decision to help you prepare a revised submission.

Summary:

In this paper, He and colleagues generated a new in vivo reporter to detect spatial and temporal activity of gene expression by combining a fast-folding destabilized fluorescent protein (FP) and a slow-folding long-lived FP. While existing transcriptional reporters combined with stable FPs are successful in detecting signaling activities, this is at the expense of detecting dynamic changes due to the long-lived fluorescent signals. The authors improved spatial and temporal resolution of these reporters by generating short-lived GFP through the inclusion of PEST peptides (GFP-PEST). Since the shorter half-life of GST-PEST caused significant loss of signals, the authors combined super-folder GFP and translational enhancing elements. The utility of these features were demonstrated by generating STAT and Notch reporters. The authors next combined short-lived GFP/long-lived RFP reporters, and generated STAT reporters that can detect dynamic STAT activity in embryos by live imaging (or in fixed sample by ratiometric imaging of the FPs). To expand this tool for broader contexts (and requiring less genetic crosses), the authors generated a clever construct they called TransTimer that includes the self-cleaving 2A peptide between edGFP and RFP under the control of a single promoter. Using TransTimer with enhancer-trapped GAL4 lines or through insertion near an enhancer region allowed the discovery of new genes showing dynamic expression changes. Indeed, the authors found that a new gene, CG30159, is required for the maintenance of intestinal stem cells. Finally, the authors generated TransTimerLex, which achieved both detection and manipulation of gene expression at the same time.

This is a nicely designed study with strong proof of principle. Together, these different transcriptional reporters enhance the current tools/methods available to study dynamic transcriptional changes. They will be very useful to the wider *Drosophila* research committee, and as pointed out by the authors. The design elements should be transferrable to other research organisms as well. In summary, we believe this is a very clever tool that will be useful for the community and *eLife* readership.

Essential revisions:

1) When Transtimer is used as a direct reporter construct under the control of a gene's promoter and enhancer, it directly reflects the expression dynamics of that gene. It is more complex when Transtimer is used as a UAS construct. This use employs GAL4 which is a stable intermediary. The authors should model how using a stable intermediary such as GAL4 affects detection of signaling dynamics. This would strengthen the conclusion that can be reached using the Transtimer as a UAS construct. If there are already GAL4 reagents (e.g. generated as part of Flylight project) for any of the genes for which the authors have a direct reporter, a head to head comparison would be beneficial.

---

## [Author Response]

Essential revisions:1) When Transtimer is used as a direct reporter construct under the control of a gene's promoter and enhancer, it directly reflects the expression dynamics of that gene. It is more complex when Transtimer is used as a UAS construct. This use employs GAL4 which is a stable intermediary. The authors should model how using a stable intermediary such as GAL4 affects detection of signaling dynamics. This would strengthen the conclusion that can be reached using the Transtimer as a UAS construct. If there are already GAL4 reagents (e.g. generated as part of Flylight project) for any of the genes for which the authors have a direct reporter, a head to head comparison would be beneficial.

We agree with the reviewer that adding the intermediary Gal4 will significantly affect the reporter dynamics. To test that, we modeled the change of reporter dynamics after adding Gal4 (Figure 6—figure supplement 1A-F), and compared the dGFP and RFP driven directly by the Notch responding element Su(H)Gbe vs. a UAS-dGFP-2A-RFP (TransTimer) driven by Su(H)Gbe-Gal4 (Figure 6—figure supplement 1G,H). The modeling result suggests that the response time (time to reduce the signal to half of the original when the promoter is off) is generally proportional to the T_G1/2_ (half-life of Gal4) + TF_P1/2_ (half-life of fluorescent protein). This suggests that adding Gal4 will generally slow down the dynamic of the reporter. This is particularly clear in Figure 6—figure supplement 1E,H: the dGFP driven directly by the enhancer is more restricted in neuroblast cells (NB) than the dGFP driven by enhancer-Gal4 due to the retention of Gal4 intermediary. However, because the Gal4/UAS can amplify the signal, the system may reveal cells that weakly express the reporter and may be missed by reporter driven directly by the enhancer: in Figure 6—figure supplement 1F,G, the pattern driven by Gal4/UAS system is significantly broader.